

**Single-particle investigation of summertime and wintertime Antarctic sea spray aerosols**
**using low-*Z* particle EPMA, Raman microspectrometry, and ATR-FTIR imaging**
**techniques**
Hyo-Jin Eom[1], Dhrubajyoti Gupta[1], Hye-Rin Cho[1], HeeJin Hwang[2], SoonDo Hur[2], Yeontae
Gim[3] and Chul-Un Ro[1,*]
[1]*Department of Chemistry, Inha University, Incheon, Republic of Korea*
[2]*Polar Climate Change Research Division, Korea Polar Research Institute, Incheon, Republic of*
*Korea*
[3]*Arctic Research Center, Korea Polar Research Institute, Incheon, Republic of Korea*
**ABSTRACT**
Two aerosol samples collected at King Sejong Korean scientific research station,
Antarctica on Dec. 9, 2011 in the austral summer (sample S1) and July 23, 2012 in the austral
winter (sample S2), when the oceanic chlorophyll-a levels were quite different, by ~19 times (2.46
vs. 0.13 µg/L, respectively), were investigated on a single particle basis using quantitative energy-
dispersive electron probe X-ray microanalysis (ED-EPMA), called low-*Z* particle EPMA, Raman
microspectrometry (RMS), and attenuated total reflectance Fourier transform infrared (ATR-FTIR)
imaging techniques to obtain their characteristics based on the elemental chemical compositions,
molecular species, and mixing state. X-ray analysis showed that the supermicron summertime and
wintertime Antarctic aerosol samples have different elemental chemical compositions, even
though all the individual particles analyzed were sea spray aerosols (SSAs); i.e., the contents of C,
O, Ca, S, and Si were more elevated, whereas Cl was more depleted, for sample S1 having a much
higher chlorophyll-a level than for sample S2. Based on qualitative analysis of the chemical species
present in individual SSAs by the combined application of RMS and ATR-FTIR imaging, different
organic species were encountered in samples S1 and S2; i.e., Mg hydrate salts of alanine were
predominant in samples S1 and S2, whereas Mg salts of fatty acids internally mixed with Mg

*Corresponding author. Tel.: +82 32 860 7676; fax: +82 32 874 9207
E-mail address: curo@inha.ac.kr (C.-U. Ro)



hydrate salts of alanine were significant in sample S2. Although $CaSO_4$ was encountered
significantly in both samples S1 and S2, other inorganic species, such as $Na_2SO_4$, $NaNO_3$,
$Mg(NO_3)_2$, $SiO_2$, and $CH_3SO_3Mg$, were encountered more significantly in sample S1, suggesting
that those compounds may be related to the higher phytoplankton activity in summer.

**INTRODUCTION**

As more than 70% of the Earth's surface is covered by ocean, sea spray aerosols (SSAs)

make a dominant contribution to the total aerosol load in the air (Quinn et al., 2015). The influence
of nascent SSAs on the Earth′s radiative balance, either directly by scattering light or indirectly by
acting as cloud droplets or ice nuclei, needs to be understood to better predict the additional
anthropogenic effects on SSAs (Ault et al., 2013a). Recently, it was suggested that SSAs mixed
with organic matter occurring at the ocean surface can have a significant influence on the Earth's
climate change (Wang et al., 2015). In addition, an understanding of the nascent SSA properties in
terms of the physical, chemical, and biological processes in the ocean surface is required to reduce
the current uncertainties for climate models (Prather et al., 2013).

SSAs are generated by bubbles bursting at the sea surface, where submicron and

supermicron SSAs are believed to be formed mostly from film drops and jet drops, respectively
(Quinn et al., 2014; Quinn et al., 2015; Wang et al., 2015). Submicron nascent SSAs were reported
to have more enriched organic species and less inorganic salts than the supermicron nascent SSAs
(Ault et al., 2013b; Prather et al., 2013; Wang et al., 2015). Although the molecular species of the
organic matter in nascent SSAs are unknown, a recent mesocosm experiment showed that
submicron SSAs were enriched with aliphatic-rich organic species, whereas supermicron SSAs
contained more oxidized organic species (Wang et al., 2015). As organic and inorganic matter in
sea water could be produced through the biological food web, the chemical compositions in
nascent SSAs would be interrelated with the biological activity in sea-water. On the other hand,
there have been disputes regarding the correlation between the biological activity in the ocean and
SSA organic matter in the marine boundary layer. Some studies reported positive correlations
between the levels of chlorophyll-a, which is an indicator of the biological activity in the sea-water,
and organic matter in SSAs (Prather et al., 2013; Hu et al., 2013; Rinaldi et al., 2010; O'Dowd et
al., 2004), whereas some showed no correlation between them (Quinn et al., 2014; Bates et al.,



2012). In addition, it was claimed that the chlorophyll-a level showed a complicated correlation
with the organic matters in nascent SSAs and the bacterial enzyme activities should also be
considered to better understand the overall generation and temporal variations of organic matter
(Wang et al., 2015), strongly suggesting the necessity for further studies.
The Antarctic region, which is isolated from anthropogenic sources, is one of the few
pristine places to study natural SSAs with a minimal anthropogenic influence (Maskey et al., 2011).
A major constituent in the Antarctic troposphere is nascent SSAs (Hara et al., 2012; Maskey et al.,
2011) and their chemical compositions can be altered through heterogeneous reactions with $SO_4^{2-}$
and $CH_3SO_3^-$ during the summer (Hara et al., 2014). Some studies on the characterization and
seasonal cycles of different aerosol species at various Antarctic locations, such as McMurdo (Ross
Island), Aboa (Queen Maud Land), Syowa (East Ongul Island), Dome Fuji (Queen Maud Land),
O'Higgins (Chile), Admiralty Bay (King George Island), and Mizuho (Atka Bay) stations, have
been carried out, and bulk and single-particle analytical techniques showed that the sea-salts and
sulfur-containing species were the most abundant constituents in Antarctic aerosol samples (Hara
et al., 2013; Hara et al., 2012; Maskey et al., 2011; Préndez et al., 2009; Biancato et al., 2006;
Hara et al., 2006; Kerminen et al., 2000; Shaw, 1988). In this study, two Antarctic aerosol samples
collected in the summer and winter, having a drastic contrast in their oceanic chlorophyll-a levels,
were characterized on a single particle basis using quantitative energy-dispersive electron probe
X-ray microanalysis (ED-EPMA), called low-$Z$ particle EPMA, Raman microspectrometry (RMS),
and attenuated total reflection Fourier transform infrared (ATR-FTIR) imaging. In the present
study, low-$Z$ particle EPMA was applied to investigate the elemental compositional contrast
between the summertime and wintertime samples and two vibrational spectroscopic techniques,
such as RMS and ATR-FTIR imaging, were employed on a single particle basis to identify the
organic and inorganic molecular species present in Antarctic aerosol samples, clearly revealing the
different chemical features between two samples.

**2. EXPERIMENTAL SECTION**
**2.1 Samples**
Aerosol samples were collected at a Korean scientific research station in the Antarctic: King
Sejong station (62°13´S, 58°47´W), which is located at King George Island, Chile (see Figure S1



of the Supporting Information). King George Island in the South Ocean is 120 km off the coast of
Antarctica, and is dominated by pervasive ice caps, with more than 90% of the island being
glaciated. This study examined two aerosol samples collected on Dec. 9, 2011 in the austral
summer (sample S1) and July 23, 2012 in the austral winter (sample S2), when the oceanic
chlorophyll-a levels were quite different, by ~19 times (2.46 vs. 0.13 μg/L, respectively, which
were determined daily for nearby oceanic water using a fluorometer (TD 700, Turner Design,
USA)). The samples were collected on Al foil substrates (Sigma-Aldrich, 99.8% purity) using a
three stage cascade impactor ($PM_{10}$ Impactor, Dekati Inc.) during daytime at temperatures, $T = 1.1$
− 2.1 °C and -1.9 − -1.5 °C and relative humidity, RH = 94.1 −  94.5 % and 87.6 − 92.1 % for
samples S1 and S2, respectively. The impactor had aerodynamic cut-sizes of 10, 2.5, and 1 μm for
stages 1, 2, and 3, respectively, at a 10 L $min^{-1}$ sampling flow, and individual particles collected
on stages 2 and 3 ($PM_{2.5-10}$ and $PM_{1.0-2.5}$ fractions with the size range of 2.5–10 μm and 1–2.5 μm,
respectively) were examined.

Three-day (72 h) backward air-mass trajectories were obtained using the Hybrid

Lagrangian Single-Particle Integrated Trajectory (HYSPLIT) model from the NOAA Air
Resources Laboratory's web server (http://www.arl.noaa.gov/ready/hysplit4.html). The back-
trajectories for samples S1 and S2 show that the air-masses at heights of 500 m, 1000 m, and 1500
m above sea level, which originated from the Pacific Ocean, travelled over the Pacific Ocean and
passed over Chile, respectively (see Figure S1 of the Supporting Information).

## 2.2 Low-$Z$ particle EPMA measurement and data analysis

Low-$Z$ particle EPMA measurements were carried out by scanning electron microscopy

(JSM-6390, JEOL) equipped with an Oxford Link SATW ultrathin window energy-dispersive X-
ray (EDX) detector, which has a spectral resolution of 133 eV for Mn Kα X-rays. The X-ray spectra
and elemental X-ray maps were recorded using INCA Energy software. An accelerating voltage of
10 kV and a beam current of 0.5 nA, and a typical measuring time of 20 s were used for the X-ray
spectral data acquisition using area mode, where the X-ray signals were obtained by the scanning
electron beam over the entire area of each particle. The net X-ray intensities for the chemical
elements were obtained by a non-linear, least-square fit of the spectra collected using the AXIL
program (Vekemans et al., 1994). The elemental concentrations of the individual particles were



determined from their X-ray intensities using a Monte Carlo calculation combined with reverse
successive approximations (Ro et al., 2003). For the X-ray mapping measurements, an accelerating
voltage and beam current are the same as the area mode measurements except for a typical
measuring time of 30 min. A more detailed discussion of the EPMA measurement conditions can
be found elsewhere (Ro et al., 2005; Ro et al., 1999).

**2.3 RMS measurements**
The particles collected on Al foil were mounted on the microscope stage of a confocal
Raman microspectrometer (XploRA, Horiba Jobin Yvon) equipped with a 100×, 0.9 numerical
aperture objective (Olympus). Raman point and mapping measurements were carried out under
ambient conditions. Optical images of the particles for relocation were obtained using a video
camera. Raman scattering was excited at the 532 nm wavelength using an air-cooled diode laser
and detected with a multichannel air cooled charge-coupled device (CCD) at an 1800 gr/mm
grating. The excitation laser power delivered to the individual particles was approximately 3 mW
using a controlled confocal hole of 300 ~ 500 μm and a slit, 100 μm in diameter. The spectral
ranges of 100–4000 cm$^{-1}$ were performed with a 5 s acquisition time and 5 times accumulation for
each measurement. The spectral resolution was 1.8 cm$^{-1}$ and the spot size of the laser beam at the
sample was estimated to be ~1 μm$^2$. The XYZ computer-controlled Raman mapping was
performed by obtaining the Raman spectra in point-by-point XY scanning mode with a 1 μm step
and a 5 s integration time per pixel. The spectra and images were acquired using Labspec6 software.
A more detailed discussion of the RMS measurement conditions for single particle analysis can be
found elsewhere (Sobanska et al., 2012; Eom et al., 2013; Jung et al., 2014).

**2.4 ATR-FTIR imaging measurements**
The ATR-FTIR imaging measurements were performed using a Perkin Elmer Spectrum
100 FTIR spectrometer interfaced to a Spectrum Spotlight 400 FTIR microscope. An ATR
accessory using a germanium hemispherical IRE crystal, 600 μm in diameter, was used for ATR
imaging. The ATR accessory was mounted on the X–Y stage of the FTIR microscope and the IRE
crystal was made to come into contact with the sample through a force lever. A spatial resolution
of 3.1 μm at 1726 cm$^{-1}$ ($\lambda = 5.79$ μm) is achievable (Van Dalen et al., 2007). A 16 x 1 pixel mercury



cadmium telluride (MCT) array detector was used to obtain the FTIR images with a pixel size of
1.56 μm. For each pixel, an ATR-FTIR spectrum, ranging from 680 to 4000 cm$^{-1}$ with a spectral
resolution of 4 cm$^{-1}$, was obtained from eight interferograms, which were co-added and Fourier-
transformed. The position of the crystal on the sample was determined using an optical microscope
equipped with a light emitting diode and a CCD camera, which allowed relocation of the same
single particles that had been analyzed using RMS before ATR-FTIR imaging. Spectral data
processing was performed using Perkin Elmer Spectrum IMAGE software. A more detailed
discussion of the ATR-FTIR imaging measurement conditions for single particle analysis can be
found elsewhere (Song et al., 2010; Song et al., 2013; Jung et al., 2014).

**3. RESULTS AND DISCUSSION**

**3.1 Single-particle characterization of the summertime and wintertime Antarctic SSAs**
**using low-$Z$ particle EPMA**

Figure 1 presents typical secondary electron images (SEIs) of the individual particles on

two PM$_{2.5-10}$ (stage 2) samples collected in the austral summer and winter, where the chemical
species comprising each particle, determined from X-ray spectral data, is indicated. All the
particles on the images are of a marine origin having major Na and Cl contents with small
quantities of C, O, Mg, K, Ca, S, and/or Si. Overall, approximately 600 particles of samples S1
and S2 examined by low-$Z$ particle EPMA were of a marine origin. Na, Mg, Cl, S, C, and O were
present in all the particles, whereas K, Ca, and Si were encountered more frequently in the
summertime sample S1 than in the wintertime sample S2 (93.6 % vs. 79.4 % encountering
frequencies for K; 93.9 % vs. 75.5 % for Ca; and 70.1 % vs. 0.7 % for Si, respectively). In particular,
Si is present exclusively in sample S1, which might be a good indicator of the phytoplankton
influence on the nascent SSAs.

As ambient relative humidity (RH) at the sampling times were higher than 87.6% and the

efflorescence RHs (ERHs) of the inorganic sea salt components (e.g., ERHs of NaCl and CaSO$_4$
are ~45-47 % and ~80-90 %, respectively (Gupta et al., 2015; Schindelholz et al., 2014; Xiao et
al., 2008)), the SSAs would be collected as aqueous droplets at the time of collection. Once



exposed at a low RH, e.g. by being either handled under the dry ambient conditions or placed in
the vacuum chamber of SEM, they would crystallize fractionally, resulting in their heterogeneous
mixing states, as shown in Figure 1, having bright and crystalline solids, segregated and somewhat
dark regions, and elongated rods (indicated by the yellow arrows in Figure 1), which are more
distinctive for the summertime particles. The fractional crystallization of SSAs has also been
reported (Ault et al., 2013a; Hara et al., 2013; Hara et al., 2014). To determine the chemical species
of the crystalline solids, dark regions, and rods, elemental X-ray and molecular Raman mapping
measurements were performed on the same individual SSA particles. Figure 2 presents the SEIs
and molecular Raman and elemental X-ray map images of two typical summertime and wintertime
SSA particles. As Raman-inactive NaCl and $MgCl_2$ species cannot generate Raman signals, Raman
mapping was performed to determine the spatial distributions of $CaSO_4$ (using Raman signal in
$1000 – 1020$ cm$^{-1}$ range), $Na_2SO_4$ (using Raman signal in $985 – 995$ cm$^{-1}$ range), and organic
species (using Raman signal in $2800 – 3000$ cm$^{-1}$ range). X-ray mapping images of Na, Mg, Ca,
Cl, S, C, and O are overlaid in different colors on the SEIs. The combined Raman and X-ray map
image data of Figure 2(a) clearly indicate that the upper bright solid (region 1, notated on the SEI
of Figure 2(a)) of the summertime SSA particle is composed of NaCl, the bottom-right region 3 is
a mixture of $MgCl_2$ and organic species (having a somewhat dark appearance due to the low
secondary electron yield of organic species), and the two elongated rods are of a mixture of $CaSO_4$
and $Na_2SO_4$. The wintertime SSA particle in Figure 2(b) is composed of NaCl (at region 1) and
the mixture of $MgCl_2$ and organic species (at region 2). As C and O are overlapping in their X-ray
maps of Figure 2, the organic species appear to contain a significant amount of oxygen. Figure 3
shows the X-ray spectra and elemental atomic concentrations obtained from the entire regions of
the summertime and wintertime particles using area-mode X-ray data acquisition. The
summertime particle contains more C, O, Si, S, and Ca than the wintertime particle. As the amount
of sulfate (by assuming all the sulfur exists as sulfate) for the summertime particle is larger than
that of Ca, the sulfate first crystallized as $CaSO_4,$ and the remaining sulfate crystallized as $Na_2SO_4$,
resulting in the formation of elongated rods composed of a mixture of $CaSO_4$ and $Na_2SO_4$. For the
wintertime particle, $CaSO_4$ was observed weakly at the upper-right region because of the low
sulfate content.

Table S1 in the Supporting Information shows the mean elemental concentrations for an



overall ~600 individual particles in $PM_{1.0-2.5}$ and $PM_{2.5-10}$ fractions of the summertime and
wintertime samples, obtained by low-$Z$ particle EPMA. As all the particles analyzed in these
samples are of a marine origin, the mean atomic concentrations of Na and Cl are largest (ranging
in 25.2 – 28.3 % and 24.8 – 29.2 %, respectively), followed by high C and O concentrations (18.8
– 27.1 % and 17.3 – 19.5 %, respectively), compared to those of Mg, Si, S, K, and Ca which are
in the range, 0.0 – 2.9 %. Based on the mean elemental weight concentrations, the C and O contents
were smaller based on the mean atomic concentrations, even though they were still considerable
(9.6 – 14.6 % and 12.0 – 13.6 %, respectively). On the other hand, the organic contents on a
molecular basis would be smaller than the elemental C contents but the molecular organic content
could not be estimated because the organic molecular species in SSAs have not been identified
clearly (Ault et al., 2013b; Laskina, 2015; Quinn et al., 2015). An interesting observation was that
all the supermicron Antarctic SSAs both in the summertime and wintertime samples were a
mixture of inorganic and organic species.
To better examine the chemical compositional contrast between samples S1 and S2, Table
1 lists the mean elemental concentration ratios to Na for individual particles together with those
for bulk sea-water. The atomic concentration ratios of C, O, Si, S, and Ca; Cl; and Mg and K for
the summertime sample were higher and lower than and similar to those of the wintertime sample,
respectively (also see Fig. S2, which clearly shows different distributions of individual particles
having specific elemental concentration ratios between the summertime and wintertime samples),
indicating that C, O, Si, S, and Ca; and Cl are enriched and depleted in the summertime sample,
respectively. In addition, those enriched and depleted elements have higher and lower
concentration ratios than the bulk sea-water ratios, respectively.
As the [C]/[Na] ratios for both samples were high compared to bulk sea-water [C]/[Na]
ratio, even the supermicron Antarctic SSAs contain significantly enriched organic species. The
[C]/[Na] ratios of sample S1 were higher than those of sample S2, suggesting that the higher
organic matter is related to the higher phytoplankton activities, and those for particles in the $PM_{1.0-2.5}$
fractions of samples S1 and S2 (1.12 and 0.83, respectively) were higher than $PM_{2.5-10}$ fractions
(0.87 and 0.70, respectively), indicating that the smaller particles contain more organic species,
which is consistent with other observations reporting more organics in the smaller SSAs (Quinn et
al., 2015).



The [O]/[Na] ratios of sample S1 are higher than those of sample S2, and those for particles

in the $PM_{2.5-10}$ fractions of samples S1 and S2 (0.77 and 0.68, respectively) are higher than the

$PM_{1.0-2.5}$ fractions (0.71 and 0.66, respectively). Similar observations were made for S and Ca, for

which the elemental concentration ratios were somewhat higher in sample S1 and in larger size

fractions (see Table 1). In addition, the frequencies of encountering particles having higher [S]/[Na]

or [Ca]/[Na] ratios than bulk sea-water were significantly higher in the summertime sample and in

the larger size fractions (see encountering frequency data for S and Ca in Table 1), indicating that

O, S, and Ca are interrelated with common sources, which is also supported by the observation of

elongated $CaSO_4$ rods in the Raman and X-ray mapping measurements. The enriched S and O in

the S1 sample appear to be due to the elevated nss-$SO_4^{2-}$ levels. In the austral summer

(November–March) of the Antarctic, higher solar radiation levels and temperatures than the other

seasons tend to enhance the phytoplankton activities (as supported by its high chlorophyll-a level

for sample S1), which enhances the production and emission of oceanic dimethyl sulfide (DMS)

(Wagenbach et al., 1998; Preunkert et al., 2008). The volatile DMS in the atmosphere undergoes

complex sequences of gas-phase oxidation reactions, generating a range of sulfur-containing

products, such as dimethyl sulfoxide (DMSO), methanesulfonic acid (MSA), $SO_2$, and $H_2SO_4$

(Gaston et al., 2010). These oxidized products can condense onto preexisting particles, resulting

in the formation of nss-$SO_4^{2-}$-containing SSAs. As $CaSO_4$ can effloresce at very high RH, the

nss-$SO_4^{2-}$ can combine easily with Ca, as observed in Figure 1, where the $CaSO_4$ rods are observed

more frequently in sample S1.

Si is encountered for the summertime particles, and more abundantly ([Si]/[Na] = 0.03 vs.

0.01) and frequently (encountering frequency = 93.4 % vs. 47.5 %) in the $PM_{1.0-2.5}$ fraction than in

the $PM_{2.5-10}$ fraction. As Si is observed mostly in sample S1 and more in the smaller size fraction,

it appears to be from fragments of silica cell walls of diatoms, a major group of algae and a

common type of phytoplankton in the oceans (Litchman and Klausmeier, 2008; Alpert et al., 2015).

In winter, the combination of reduced diatom activities and enhanced sea ice (Kurahashi-

Nakamura et al., 2007; Vancoppenolle et al., 2013) would hinder the emission of Si species into

the atmosphere, resulting in the scarce observation of Si in the S2 sample.

In the SSAs of samples S1 and S2, only Cl is depleted compared to bulk sea-water ([Cl]/[Na]

= 1.00 and 1.03 for samples S1 and S2, respectively, vs. 1.16 for sea-water), and the Cl depletion



is somewhat higher for the summertime SSAs than the wintertime and for $PM_{1.0-2.5}$ fractions than
$PM_{2.5-10}$ fractions, suggesting that Cl was liberated by the reactions of NaCl and/or $MgCl_2$ with
$nss-SO_4^{2-}$ and/or $CH_3SO_3^-$, which are more abundant in the summer, with more depletion for
smaller SSAs having a higher surface to volume ratio and higher reactivity.

**3.2 Single-particle molecular speciation of Antarctic SSAs using RMS and ATR-FTIR**
**imaging**

Based on low-$Z$ particle EPMA analysis, the C, O, Si, S, and Ca levels were elevated for
the summertime SSAs on a single-particle basis. This quantitative elemental X-ray analysis
provides useful information on their morphology, elemental chemical compositions, and mixing
states of individual Antarctica SSAs. On the other hand, as low-$Z$ particle EPMA has a limitation
on molecular speciation and hydrogen detection, the RMS and ATR-FTIR imaging techniques
were applied in combination for the analysis of the same individual SSAs to investigate their
Raman- and IR-active organic and inorganic molecular species. Raman and ATR-FTIR techniques
are useful because their spectra of organic and inorganic compounds are quite specific depending
on their chemical species, phase, crystallinity, and neighboring environment. In particular, the
complicated vibrational spectral patterns observed in the fingerprint region (< 1500 $cm^{-1}$) in the
Raman and FTIR spectra can be critically useful for the positive or negative identification of
specific organic compounds with the same phase and crystallinity. In addition, the differences in
their spectra owing to their different signal generation mechanisms (i.e., scattering vs. absorption
of energy) and different selection rules would make the two fingerprint techniques rather
complementary (Jung et al., 2014).

**3.2.1 Organic species**
Among the ~250 individual SSAs of samples S1 and S2 investigated by RMS and ATR-
FTIR imaging techniques, the frequently encountered organic species are Mg hydrate salts of
alanine (MgAla) and Mg salts of fatty acids (MgFAs).
Figure 4 shows baseline-corrected Raman and ATR-FTIR spectra of two individual
summertime SSAs containing mainly two types of MgAla (detailed identification is given later.)



with some inorganic compounds. If several peaks from inorganic compounds (i.e., Raman peaks
at 124 and 467 $cm^{-1}$ for $SiO_2$, at 717 and 1052 $cm^{-1}$ for $Mg(NO_3)_2$, at 989 $cm^{-1}$ for $Na_2SO_4$, at 1008
$cm^{-1}$ for $CaSO_4 \cdot 2H_2O$, and at 1068 $cm^{-1}$ for $NaNO_3$; and ATR-FTIR peaks at 1087 and 1165 $cm^{-1}$
for $SiO_2$, at 1100 $cm^{-1}$ for $Na_2SO_4$ and $CaSO_4 \cdot 2H_2O$) are excluded from the consideration, the
Raman and ATR-FTIR spectra of two types of SSAs are similar except for their different Raman
and ATR-FTIR peak shapes. That is, the Raman peaks of crystalline water are sharp at 3276 and
3390 $cm^{-1}$ for Type 1 SSA, compared to the relatively broad peak at 3410 $cm^{-1}$ for Type 2 SSA.
The C-H vibration Raman peaks of Type 1 SSA are split at 3000/2988 $cm^{-1}$ and 2940/2919 $cm^{-1}$,
which correspond to the non-split Raman peaks of Type 2 SSA at 2989 $cm^{-1}$ and 2939 $cm^{-1}$. The
C-H bending Raman peaks of Type 1 SSA are split into 1433/1457/1479 $cm^{-1}$, which correspond
to the Raman peaks of Type 2 SSA at 1427/1452 $cm^{-1}$. In the fingerprint region, the characteristic
Raman peaks both for Types 1 and 2 SSAs are observed at 869, 1102, 1130, 1254, ~1300, ~1370,
and 1640 $cm^{-1}$. Similarly, the ATR-FTIR peaks of crystal water are sharp and broad at 3265 and
3370 $cm^{-1}$ for Type 1 SSA and 3372 $cm^{-1}$ for Type 2 SSA, respectively, even though the C-H
vibration ATR-FTIR peaks are unclear for both types of SSAs. In the ATR-FTIR spectra, the water
bending peaks at ~1640 $cm^{-1}$ are quite strong for both Types SSAs with the peak of Type 1 SSA
being much sharper. In the fingerprint region, the characteristic ATR-FTIR peaks for both Types 1
and 2 SSAs at 770, 869, 1127, 1254, 1312, ~1360, 1376, 1428, 1476, and 1507 $cm^{-1}$ were sharp
and broad for the Type 1 and Type 2 SSAs, respectively. Similar Raman and ATR-FTIR peak
patterns of the Types 1 and 2 spectra except for their different peak shapes strongly indicates that
they have the same in chemical compositions but with different crystal structures. As amorphous
solids tend to provide broader Raman and ATR-FTIR peaks than crystalline solids (Shebanova and
Lazor, 2003; Gouadec and Colomban, 2007; Lutz and Haeuseler, 1999; Yan et al., 2008), the Types
1 and 2 SSAs are most likely amorphous and crystalline solid particles, respectively.

Figure S3 shows the Raman and ATR-FTIR spectra of aerosols generated by the

nebulization of a mixture solution of 0.2 M alanine and 0.1 M $MgCl_2$ standard chemicals and
collected on Al foil. All the fresh aerosol particles immediately after nebulization showed the first
pair of Raman and ATR-FTIR spectra in Figure S3 on a single particle basis, which resemble the
Raman and ATR-FTIR spectra shown in Figure 4(b) when the Raman and ATR-FTIR peaks from
the inorganic compounds are excluded. In particular, the ATR-FTIR spectra in Figure 4(b) and





Figure S3 appear similar. When the aerosols were measured ~1 year later after the generated
aerosols had been sealed in a plastic box and stored in a desiccator, approximately half of the
generated aerosols showed a second pair of Raman and ATR-FTIR spectra, as shown in Figure S3,
and the other half showed a third pair. The third spectra pair appears similar to those in Figure 4(a)
for a crystalline solid SSA, whereas the second spectra pair appears to be between the first and
third spectra pairs in Figure S3, strongly suggesting that the fresh aerosols generated from the
alanine and $MgCl_2$ solution are a somewhat amorphous form of MgAla, whereas the second and
third spectra pairs suggest a more crystalline nature of MgAla. The Raman peaks of the aerosols
generated at 3409 and 1637 $cm^{-1}$ are not from free water because these Raman peaks were
unchanged even at very low RH (< 5%) when the in-situ Raman measurement was performed by
changing the RH in the hygroscopic measurement system described in a previous study (Gupta et
al., 2015). This means that the intensities and shapes of the Raman peaks should be reduced and
changed, respectively, when the RH is decreased to a very low level if these peaks are from free
water. In other words, the peaks are from the hydrate crystal water bound for divalent Mg
compounds as the narrow peak shapes and peak positions resemble those of the known spectra of
$MgCl_2 \cdot 6H_2O$ and $MgCl_2 \cdot 4H_2O$ solids with hydrate crystal water (Gupta et al., 2015).

Based on a comparison of the Raman and ATR-FTIR spectra obtained for the summertime

SSAs and aerosols generated from the mixture solution of standard alanine and $MgCl_2$, the organic
species are most probably the Mg hydrate salts of alanine (MgAla), even though the precise
molecular formula could not be confirmed. The Raman spectrum, which is the same as that of
crystalline MgAla, was also observed for nascent SSAs produced using breaking waves, even
though their molecular species were not identified (Ault et al., 2013b; Wang et al., 2015). In a
previous study, the ATR-FTIR spectra were obtained from other summertime Antarctica SSAs,
which appear very similar to that of amorphous MgAla (Maskey et al., 2011). Interestingly, almost
all the ATR-FTIR spectra obtained in the previous work were for amorphous MgAla, whereas
among the 254 individual SSAs analyzed in this study, the number of crystalline and amorphous
MgAla-containing SSAs were 246 and 8, respectively, based on their Raman and ATR-FTIR
spectra. How crystallization from SSAs occurred to form these organic Mg hydrate salts in the
Antarctic environment is unclear because crystalline salts could not be made under very dry
conditions and by oven-drying overnight. On the other hand, crystalline salts were encountered





from the generated aerosol sample stored for ~ 1 year in a desiccator. Some efficient efflorescence
seeds should be present in the Antarctic SSAs, which have much more complicated chemical
compositions than the mixture solution of pure alanine and $MgCl_2$. The identification of an
accurate molecular formula and structure of MgAla and an investigation of the crystallization
mechanism requires further study.

The dominant dissolved amino acid in sea-water is glycine followed by alanine, aspartic

acid or serine (Ogawa and Tanoue, 2003; Dittmar et al., 2001). In sea-water, MgAla species would
be present as dissolved organic matter (DOM) in the form of alanine before being airborne. On the
other hand, based on the Raman and ATR-FTIR spectra of standard powdery glycine and aerosol
particles nebulized from aqueous mixtures of glycine/$MgCl_2$ and glycine/alanine/$MgCl_2$ as well as
other common target chemicals for organic matter in nascent SSAs such as sodium dodecyl sulfate,
a dipeptide of alanine and glycine, a polypeptide, and lipopolysaccharides, which are shown in
Figure S4, it is clear that MgAla-containing SSAs are composed of almost pure alanine without
glycine and others. As the Raman and ATR-FTIR sensitivities for alanine and glycine are
comparable and the same Raman spectrum for MgAla was also observed in the nascent SSAs
produced from breaking waves, there must be some unknown processes for the generation of
MgAla SSAs from sea-water.

Figure 5 shows the baseline-corrected Raman and ATR-FTIR spectra of two individual

SSAs of sample S2 containing mainly MgFAs and both MgAla and MgFAs. As shown in Figure
S5, the Raman spectra of powdery standard Mg palmitate, palmitic acid, Mg stearate, and stearic
acid appear similar except for minor differences in relative peak intensities, which is not sufficient
to identify the organic species having the Raman spectrum of Figure 5(a). On the other hand, Mg
palmitate/stearate and palmitic/stearic acids have very different ATR-FTIR spectra as shown in
Figure S5. Owing to their additional strong peaks at ~1700 $cm^{-1}$ for the -COOH functional group
and very different peak patterns in the fingerprint region of $700 - 1600$ $cm^{-1}$, palmitic/stearic acids
can be clearly distinguishable from Mg palmitate/stearate. The ATR-FTIR spectrum of Mg
palmitate is different from that of Mg stearate based on the strong hydrate peaks at 3374 and 3256
$cm^{-1}$ for Mg palmitate and the clearly different peak patterns in the wavenumber range, 1200 -
1600 $cm^{-1}$, between those of Mg palmitate and stearate. Figure S6 shows the ATR-FTIR spectra of
Mg palmitate, Mg stearate, a mixture of Mg palmitate and stearate (by 3:1), and MgFAs-containing



SSA, where the spectra of the mixture particle and the SSA match quite well, indicating that the
exemplar Antarctic SSA is a mixture of Mg palmitate and stearate. Therefore, this type of SSA is
called the Mg salts of fatty acids (MgFAs) above. The same Raman spectrum as that of MgFAs
was also observed for the nascent SSAs produced using breaking waves (Ault et al., 2013b; Wang
et al., 2015). As the pKa of palmitic and steric acids is 4.95, the palmitic/stearic acid moieties
degraded from the lipids would exist predominantly as surfactant palmitate/stearate in SSML
and/or on sea-surface and would crystallize as their Mg salts after the MgFAs-containing SSAs
were airborne by bubble busting.

**3.2.2 Inorganic species**

The Raman and IR active inorganic species encountered in the Antarctic SSAs were $CaSO_4$,
$Na_2SO_4$, $NaNO_3$, $Mg(NO_3)_2$, $NH_4NO_3$, $CH_3SO_3Mg$ (Mg methanesulfonate), and $SiO_2$ and their
standard Raman and ATR-FTIR spectra are shown in Figure S7. The inorganic species present in
the SSAs could be identified clearly by matching both the Raman and ATR-FTIR spectra of the
SSAs with those of the standard inorganic compounds, even though the inorganic species in the
SSAs were observed together with organic species so that the Raman and ATR-FTIR peaks of
inorganic species sometimes appear weak compared to those of organic species. On the other hand,
even under that situation, RMS is a powerful tool as the Raman peaks of inorganic compounds are
quite useful for identifying them.

**3.3 Single-particle characterization of Antarctic SSAs using RMS and ATR-FTIR imaging**

Table 2 shows relative encountering frequencies of the organic and inorganic species for
~250 individual Antarctic SSAs. The encounter frequency of certain chemical species was
determined by counting the number of individual SSAs containing the species, regardless of its
content as the Raman and ATR-FTIR spectral data were used for qualitative molecular speciation.
Based on X-ray analysis, C and O were present in all the analyzed Antarctic SSAs. Indeed, organic
salt species were detected for all the particles of samples S1 and S2, showing that organic species
are ubiquitously present, even in supermicron SSAs. As shown in Table 2, organic salt species



were categorized into three groups containing (i) MgAla, (ii) MgFAs, and (iii) mixtures of the two
organic salts. The Raman and IR active inorganic salts were always observed together with organic
salt species, so that the relative encountering frequencies of inorganic species are shown in each
organic group.

All the particles of sample S1 contained only MgAla together with other inorganic species.

In particular, $CaSO_4$ and $Na_2SO_4$ are mixed almost internally with MgAla (for $PM_{1.0-2.5}$ and $PM_{2.5-10}$
fractions, the encountering frequencies of $CaSO_4$ were 98.3% and 92.9%, respectively, and those
of $Na_2SO_4$ were 98.3% and 88.6%, respectively), indicating that $SO_4^{2-}$ is mostly in the form of a
$CaSO_4$ and $Na_2SO_4$ mixture. For the $PM_{1.0-2.5}$ and $PM_{2.5-10}$ fractions, the overall encountering
frequencies of $Mg(NO_3)_2$ are 51.7% and 77.1%, respectively, and those of $NaNO_3$ were 0.0% and
38.6%, respectively, where the $NO_3^-$ moiety was observed more in the $PM_{2.5-10}$ fraction. The reason
for why the $NO_3^-$ moiety is more abundant in the $PM_{2.5-10}$ fraction is unclear. The $SiO_2$
concentration was 46.6% and 27.1% in the $PM_{1.0-2.5}$ and $PM_{2.5-10}$ fractions, respectively. $SiO_2$
appears to be in colloidal form because $SiO_2$ species are not water-soluble and were observed more
in the $PM_{1.0-2.5}$ fraction than in $PM_{2.5-10}$. A small number of Mg methanesulfonate was observed
only in the $PM_{1.0-2.5}$ fraction of sample S1. Higher phytoplankton activities in the summer enhance
the production and emission of oceanic DMS, resulting in the production of MSA, which is a
strong acid that can exist as an anion in sea-water and is observed as Mg salts in SSAs, even though
its encountering frequency is not high compared to other sulfates.

A significant portion of SSAs of sample S2 contain only MgAla (overall 76.6% and 33.9%

for $PM_{1.0-2.5}$ and $PM_{2.5-10}$ fractions, respectively) (see Table 2). Considering the encountering
frequencies of MgAla mixed with MgFAs (21.9% and 54.8% for $PM_{1.0-2.5}$ and $PM_{2.5-10}$ fractions,
respectively), MgAla is also almost ubiquitous in sample S2 (overall 98.5% and 88.7% for $PM_{1.0-2.5}$
and $PM_{2.5-10}$ fractions, respectively). MgFAs mixed internally with MgAla was encountered
significantly in sample S2 (overall 23.5% and 66.1% for $PM_{1.0-2.5}$ and $PM_{2.5-10}$ fractions,
respectively). For the $PM_{1.0-2.5}$ and $PM_{2.5-10}$ fractions, the encountering frequencies of $CaSO_4$ were
98.5% and 88.6% overall, respectively, whereas those of $Na_2SO_4$ were 26.6% and 8.0%,
respectively, indicating that $SO_4^{2-}$ is mostly in the form of $CaSO_4$. For the $PM_{1.0-2.5}$ and $PM_{2.5-10}$
fractions, the overall encountering frequencies of $Mg(NO_3)_2$ were 43.8% and 75.8%, respectively,
and those of $NaNO_3$ were 12.5% and 27.4%, respectively, where the $NO_3^-$ moiety was also



observed more in the PM$_{2.5-10}$ fraction. SiO$_2$ was encountered much less frequently, 7.9% and 3.2%
in the PM$_{1.0-2.5}$ and PM$_{2.5-10}$ fractions, respectively, compared to those of sample S1 (i.e., 46.6%
and 27.1%, respectively). The observation of a higher encountering frequency of SiO$_2$ in sample
S1 is consistent with that of X-ray analysis, where the detection of the Si X-ray signal was 70.1 %
and. 0.7 % for samples S1 and S2, respectively.

The relative encountering frequency data for the organic and inorganic species of samples

S1 and S2 clearly show their different chemical compositional features. MgAla is predominant for
samples S1 and S2. The MgFAs were not encountered in sample S1, but were encountered in
sample S2, mostly as internal mixtures with MgAla. As alanine is water-soluble and anions of fatty
acids are surfactants, they would be present mostly at the bulk sea-water and SSML/sea-surface,
respectively, before becoming airborne. Therefore, alanine- and fatty acids-containing SSAs are
expected to be airborne through jet- and film-drop production during bubble busting, resulting in
the generation of supermicron and submicron SSAs, respectively (de Leeuw et al., 2011; Quinn et
al., 2015). In this study, supermicron SSAs were investigated for which MgAla is almost
ubiquitous in samples S1 and S2, indicating that the supermicron SSAs were generated as jet-drops.
As MgFAs was observed mostly together with MgAla in sample S2, the MgFAs-containing SSAs
originating from film-drops might agglomerate with MgAla-containing supermicron SSAs in the
air.

In a recent mesocosm experiment, the organic matter in SSAs generated from the wave

braking of natural sea-water was monitored for 29 days after adding nutrients at the beginning of
the experiment during which two phytoplankton blooms occurred (Wang et al., 2015). The
aliphatic-rich organic matter level in the nascent SSAs was enhanced during the first bloom,
whereas the oxygen-rich organic matter level increased at the early period of the experiment before
the first bloom and remained somewhat constant thereafter, including the second bloom period.
The MgAla and MgFAs observed in this study are the aliphatic-rich and oxygen-rich organic
matters in their work, respectively, because the Raman spectra of MaAla and MgFAs are the same
as those for oxygen- and aliphatic-rich organic matters and the O/C atomic ratios of alanine,
palmitic, and stearic acids are 0.67, 0.13, and 0.11 (in their work, O/C > 0.5 for oxygen-rich organic
matters and <0.25 for aliphatic-rich organic matters). In this study, the summertime Antarctic SSAs
contain oxygen-rich organic matter, such as MgAla, whereas the wintertime SSAs contain





aliphatic-rich organic matter, such as MgFAs as well as oxygen-rich organic matter. The aliphatic-
rich organic matter was observed only during the first bloom in the mesocosm experiment, whereas
supermicron MgFAs-containing SSAs were encountered only in the wintertime sample S2
collected during no bloom event, suggesting that the chemical features of organic matter in nascent
SSAs cannot be correlated consistently with the phytoplankton activity. As microalgae can produce
more lipid and less protein under environmental stress, such as limited nutrients and low
temperature (Wu et al., 2011; Yu et al., 2009; Olson and Ingram, 1975), MgFAs, which were
biodegraded from lipid, may be observed more frequently in the wintertime oligotrophic Antarctic
Ocean with a lower temperature.

**4. Summary**

X-ray analysis of a single particle analysis showed that the supermicron summertime and

wintertime Antarctic samples have different elemental chemical compositions, even though all the
individual particles analyzed were SSAs; i.e., contents of C, O, Ca, S, and Si are more elevated,
whereas Cl is more depleted, for the summertime sample S1 with a much higher chlorophyll-a
level than for the wintertime sample S2. The combined application of RMS and ATR-FTIR
imaging to the same individual SSAs made the molecular speciation of the encountered organic
and inorganic species feasible. Based on qualitative analysis of the chemical species present in the
individual SSAs by RMS and ATR-FTIR imaging, different organic species were encountered in
samples S1 and S2, i.e., Mg hydrate salts of alanine is predominant in the S1 and S2 samples,
whereas Mg salts of fatty acids mixed internally with Mg hydrate salts of alanine are significant
in sample S2. Although $CaSO_4$ are encountered significantly in both samples S1 and S2, the other
inorganic species, such as $Na_2SO_4$, $NaNO_3$, $Mg(NO_3)_2$, $SiO_2$, and $CH_3SO_3Mg$ were encountered
more significantly in sample S1, suggesting that they reflect the high phytoplankton activity in the
summer.

In this study, there were new observations regarding the chemical compositional features

of nascent Antarctic SSAs and some of them need to be explained in further studies. First, although
just two SSA samples collected in the summer and winter were investigated, their chemical
compositional features were clearly different in terms of their chemical species and/or levels of
inorganic and organic moieties, which are related to their different oceanic biological environments





suggested by the drastic chlorophyll-a level contrast. Second, even the supermicron SSAs were
enriched significantly by organic matter, and thus the effects of organic matter in supermicron
SSAs need to be considered more seriously in a radiative forcing model study. Third, based on the
Raman and ATR-FTIR measurements, the organic moieties in SSAs are believed to be present as
the salt forms of surprisingly simple organic compounds, such as alanine and palmitic/stearic acids,
which appear to be the biodegraded final products from proteins and lipids, respectively, but the
reason for why alanine and palmitic/stearic acids are predominant as the final products is unclear.
In addition, the Mg hydrate salts of alanine are almost ubiquitous in both the summertime and
wintertime supermicron SSAs but the Mg salts of fatty acids were encountered only in the
wintertime supermicron SSAs, which will require further study to better understand the generation
processes of Antarctic SSAs.

**Acknowledgements**
This study was supported by Basic Science Research Programs through the National Research
Foundation of Korea (NRF) funded by the Ministry of Education, Science, and Technology
(NRF-2015R1A2A1A09003573).

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





Table 1. Atomic concentration ratios of the chemical elements to Na for individual particles in the summertime and wintertime $PM_{2.5-10}$ and $PM_{1.0-2.5}$ fractions.

| sample | | Summertime sample S1 | | Wintertime sample S2 | |
|---|---|---|---|---|---|
| size fraction | | $PM_{1.0-2.5}$ (stage 3) | $PM_{2.5-10}$ (stage 2) | $PM_{1.0-2.5}$ (stage 3) | $PM_{2.5-10}$ (stage 2) |
| number of particles analyzed | | 146 | 148 | 154 | 156 |
| average size (μm) | | 2.0 (±0.6) | 2.9 (±1.5) | 1.7 (±0.8) | 3.2 (±1.5) |
| Elemental ratios | Sea-water ratios in atomic conc.* | Atomic concentration ratios | | | |
| [C]/[Na] | 0.01 | 1.12 (±0.35) | 0.87 (±0.33) | 0.83 (±0.33) | 0.70 (±0.24) |
| [O]/[Na] | 114.03** | 0.71 (±0.23) | 0.77 (±0.25) | 0.66 (±0.22) | 0.68 (±0.24) |
| [Mg]/[Na] | 0.11 | 0.09 (±0.02) | 0.11 (±0.04) | 0.11 (±0.03) | 0.10 (±0.03) |
| [Cl]/[Na] | 1.16 | 0.98 (±0.05) | 1.01 (±0.05) | 1.02 (±0.10) | 1.04 (±0.04) |
| [K]/[Na] | 0.02 | 0.02 (±0.01) | 0.02 (±0.01) | 0.01 (±0.01) | 0.02 (±0.01) |
| [S]/[Na] | 0.06 | 0.065 (±0.015) | 0.070 (±0.019) | 0.058 (±0.013) | 0.059 (±0.016) |
| [Ca]/[Na] | 0.02 | 0.022 (±0.009) | 0.027 (±0.011) | 0.018 (±0.029) | 0.023 (±0.012) |
| [Si]/[Na] | 0.00 | 0.03 (±0.02) | 0.01 (±0.01) | 0.00 | 0.00 |
| Encountering frequency of particles with [S]/[Na] > 0.06 | | 52.7% | 69.7% | 43.8% | 41.9% |
| Encountering frequency of particles with [Ca]/[Na] > 0.02 | | 48.4% | 69.7% | 31.5% | 48.6% |
| Encountering frequency of particles with [Si]/[Na] > 0.00 | | 93.4% | 47.5% | - | - |
| * refs. : Haynes, W. M., 2015; Hara et al., 2005 ** [O]/[Na] value for sea-water is not meaningful as $H_2O$ content in sea-water is considered. | | | | | |





Table 2. Relative encountering frequencies (in %) of the organic and inorganic species of individual summertime and wintertime SSAs.

| Organic salt group | sample | Summertime sample S1 | | Wintertime sample S2 | |
|---|---|---|---|---|---|
| | size fraction | $PM_{1.0-2.5}$ (stage 3) | $PM_{2.5-10}$ (stage 2) | $PM_{1.0-2.5}$ (stage 3) | $PM_{2.5-10}$ (stage 2) |
| | number of particles analyzed | 58 | 70 | 64 | 62 |
| containing Mg hydrate salts of alanine (MgAla) | *overall* | *100.0* | *100.0* | *76.6* | *33.9* |
| | with $CaSO_4$ | 98.3 | 92.9 | 76.6 | 29.0 |
| | with $Na_2SO_4$ | 98.3 | 88.6 | 18.8 | 4.8 |
| | with $Mg(NO_3)_2$ | 51.7 | 77.1 | 32.8 | 22.6 |
| | with $NH_4NO_3$ | 3.4 | - | 6.3 | - |
| | with $NaNO_3$ | - | 38.6 | 7.8 | 14.5 |
| | with Mg methanesulfonate | 3.4 | | | |
| | with $SiO_2$ | 46.6 | 27.1 | 6.3 | 1.6 |
| containing Mg salts of fatty acids (MgFAs) | *overall* | | | *1.6* | *11.3* |
| | with $CaSO_4$ | | | 1.6 | 6.4 |
| | with $Mg(NO_3)_2$ | | | 1.6 | 3.2 |
| containing both MgAla and MgFAs | *overall* | | | *21.9* | *54.8* |
| | with $CaSO_4$ | | | 20.3 | 53.2 |
| | with $Na_2SO_4$ | | | 7.8 | 3.2 |
| | with $Mg(NO_3)_2$ | | | 9.4 | 50.0 |
| | with $NH_4NO_3$ | | | 1.6 | - |
| | with $NaNO_3$ | | | 4.7 | 12.9 |
| | with $SiO_2$ | | | 1.6 | 1.6 |





Figure 1. Typical secondary electron images (SEIs) of aerosol particles on stages 2 of the austral (a) summertime and (b) wintertime samples collected at King Sejong station, Antarctica.

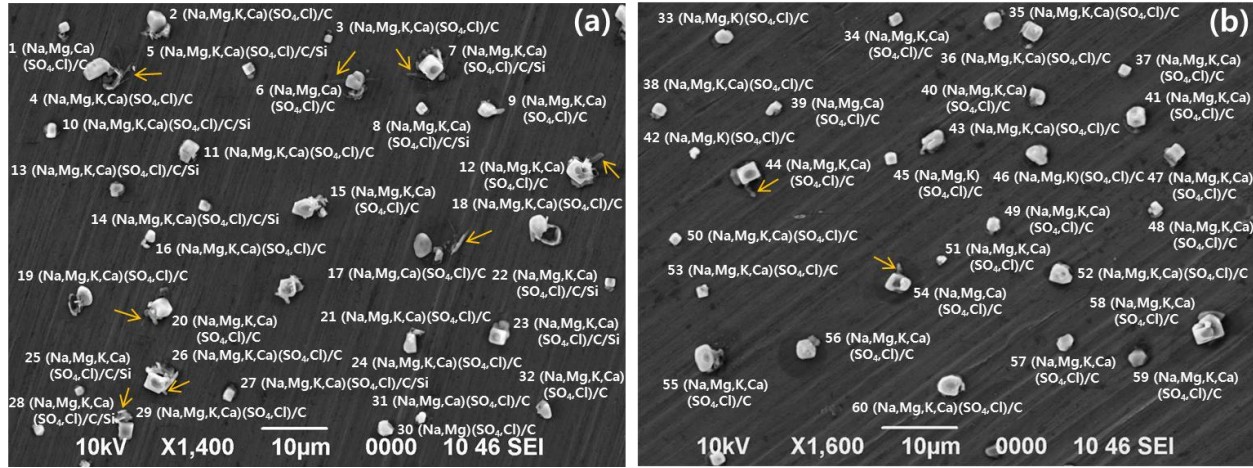



Figure 2. Secondary electron, molecular Raman map, and elemental X-ray map (overlaid on SEIs) images of two typical (a) summertime and (b) wintertime SSAs.

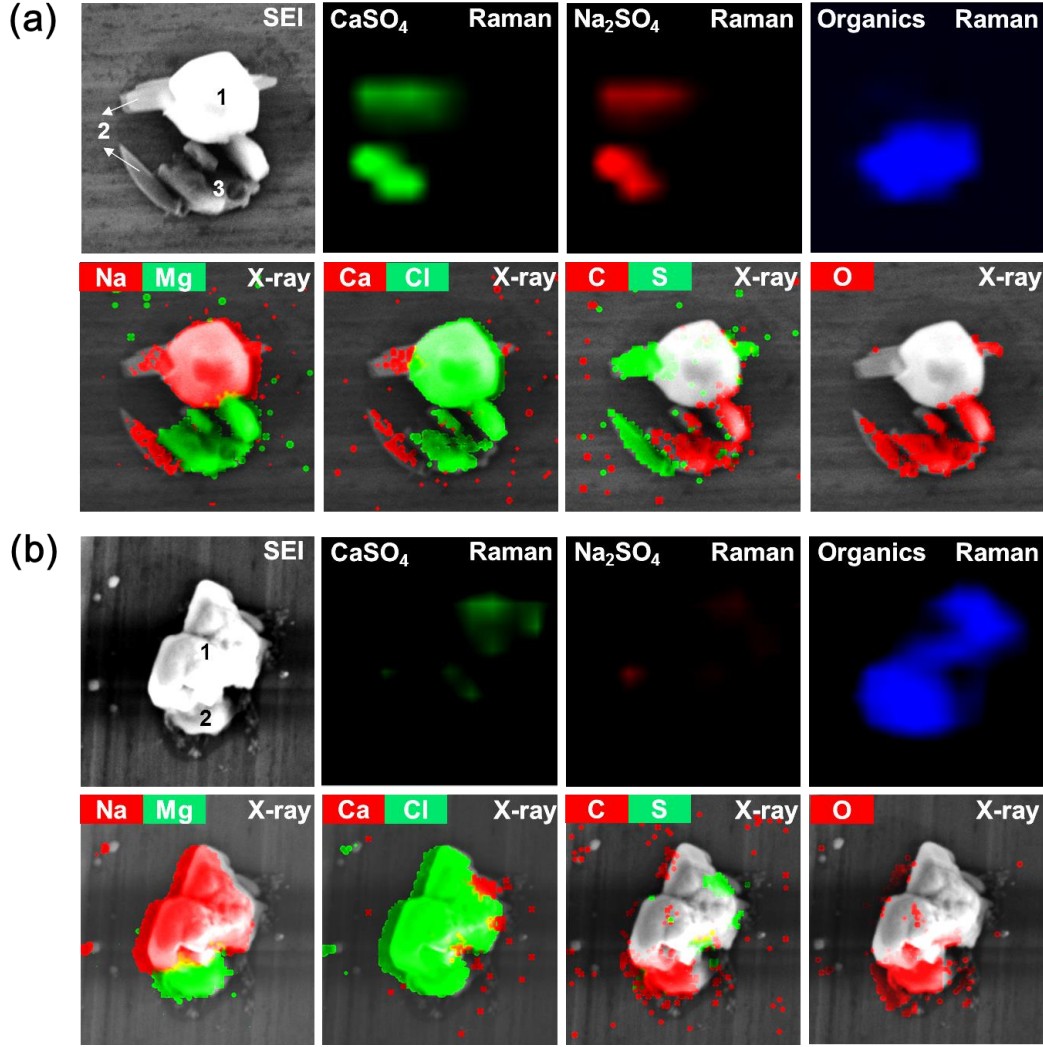





Figure 3. X-ray spectra and elemental atomic concentrations (in atomic %) of (a) the summertime
and (b) wintertime SSA particles shown in Figure 2.

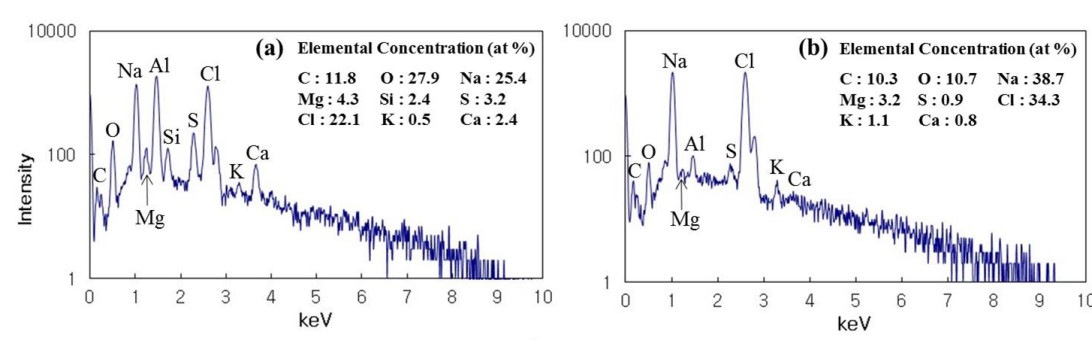




Figure 4. Raman and ATR-FTIR spectra of two typical individual summertime SSAs. The ATR-FTIR data from the 2200–2390 cm$^{-1}$ region, where the atmospheric $CO_2$ peaks are present, were deleted for clarity.

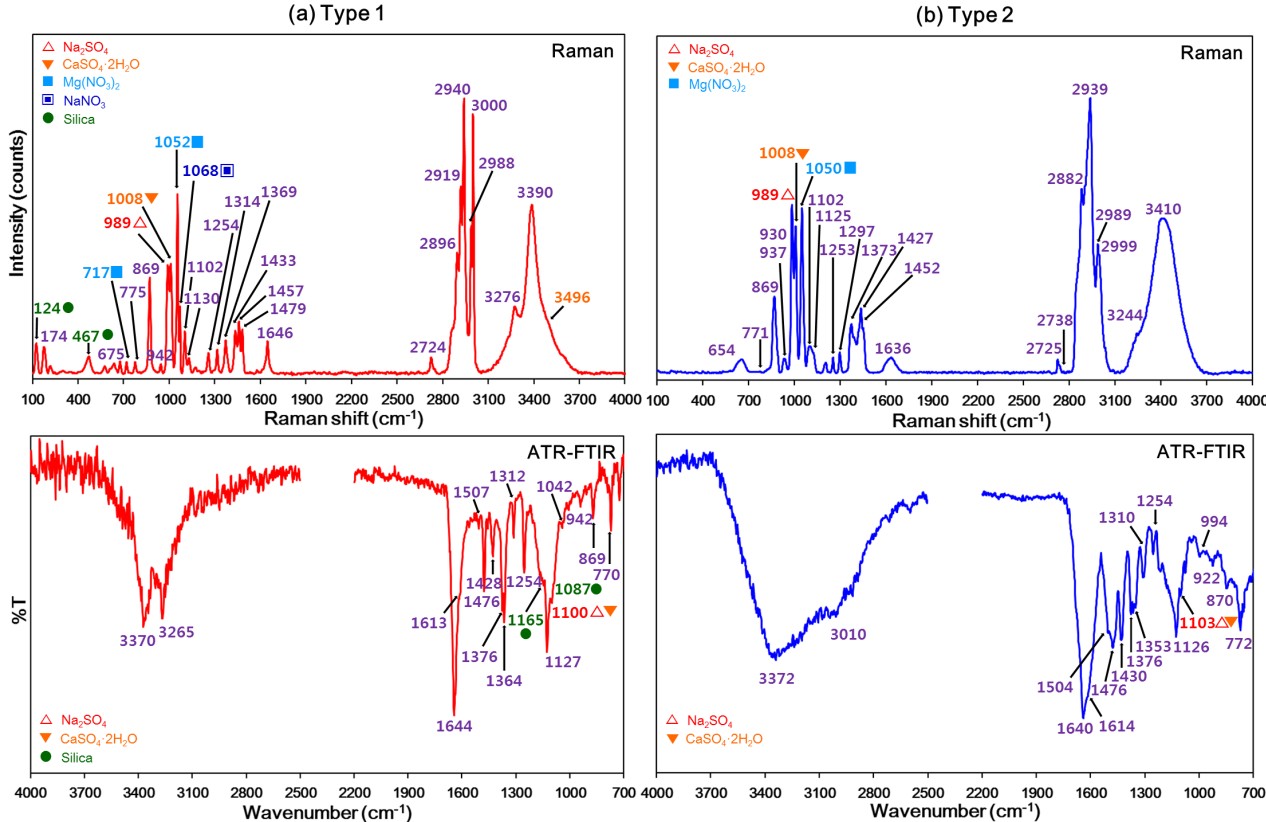



Figure 5. Raman and ATR-FTIR spectra of two typical individual wintertime SSAs. The ATR-FTIR data from the 2200–2390 cm$^{-1}$ region, where atmospheric $CO_2$ peaks are present, were deleted for clarity.

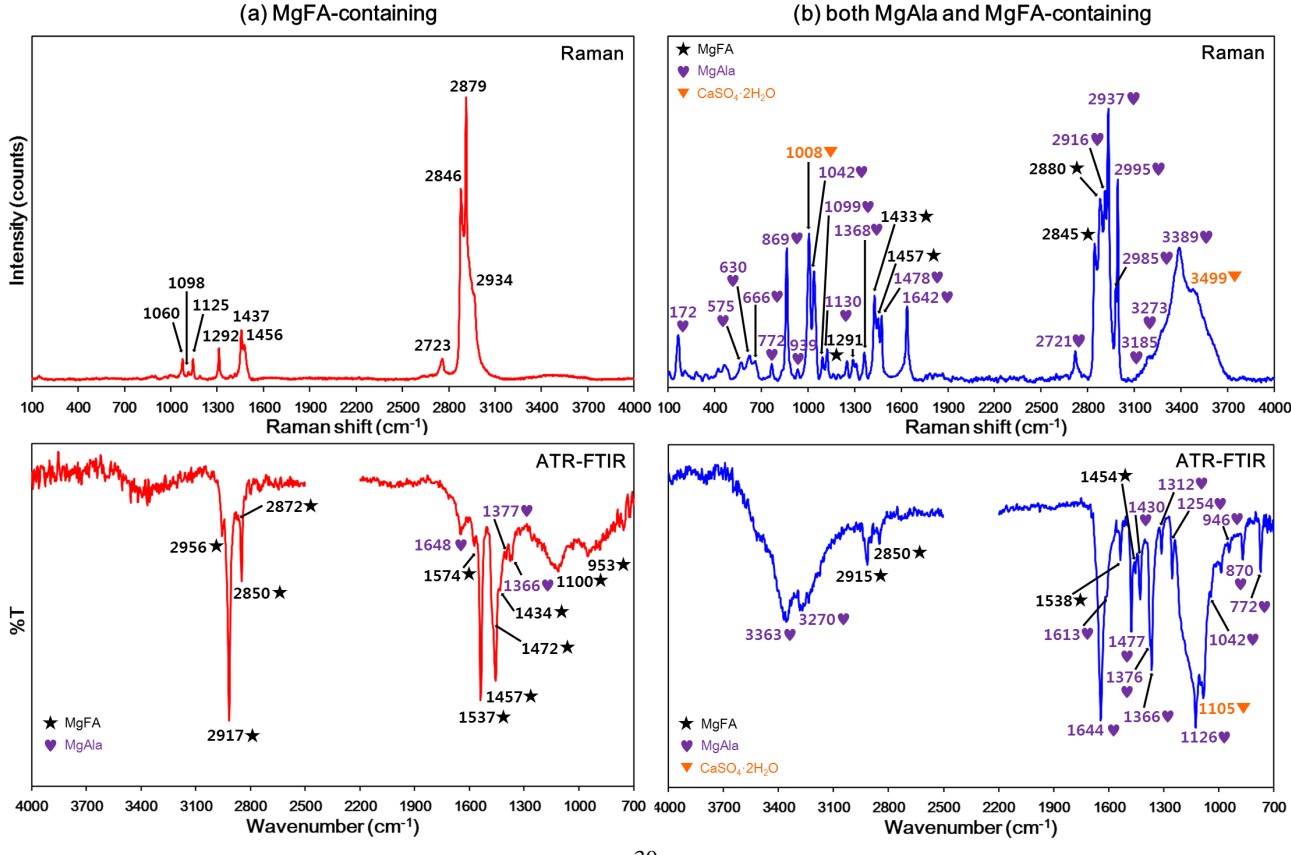