# Peer review of "Single-particle investigation of summertime and wintertime Antarctic sea spray aerosols"

_Atmospheric Chemistry and Physics, 2016_

## Referee Comment (RC1) · Anonymous Referee #2 · 3 Oct 2016

This is an interesting paper that focuses on the integration of several single particle techniques to investigate sea spray aerosol samples collected at King Sejong Korean scientific research station in the austral summer and in the austral winter. Through use of these single particle techniques several conclusions are drawn related to the identification of organic compounds and inorganic salts. Some of the observations reported in the paper have been seen before and confirm earlier studies (e.g. chloride depletion in particles).

Some questions include:

1. In many particles, there is nitrate observed in the particles - where does the nitrate come from in these particles?

2. Why is alanine such a dominant factor in the SSA? Are there other compounds that can have similar spectral features? It seems too simple to have one compound and one complex Mg-alanine in sea spray particles. The case for fatty acids seems more convincing given several studies that have identified palmitate and stearate in sea spray aerosol.

3. Some of the figure captions in the supplemental do not match or explain well the figure making it difficult to understand what is being shown. (See for example Figure S3 – what are the three sets of spectra shown?)

Overall an interesting paper, but I am concerned in some cases of over interpreting the data (e.g. point 2 above).

---

## Referee Comment (RC2) · Anonymous Referee #1 · 7 Oct 2016

**Review of "Single particle investigation of summertime and wintertime Antarctic sea spray aerosols using low-Z particle EPMA, Raman microspectroscopy, and ATR-FTIR imaging techniques."**

**Overview**

This manuscript is a very solid description of aerosols from a unique and understudied region (Antarctic). The study provides unique and valuable data regarding two samples (summer and winter) that have distinct differences in their composition. The combination of the three methods, particularly the Raman/ATR-FTIR comparison, is a powerful unified approach to explore the detailed physicochemical composition of these particles. The Ro group has really advanced the ATR-FTIR coupled with Raman approach that allows for insights regarding how sulfate is bound, for example. Overall, the data interpretation is thorough and solid. The authors do a nice job of placing their results in the context of other research in the field. A few small issues should be addressed in a revised manuscript and are indicated below. Overall, this manuscript has the potential to be a useful contribution to the field.

**Major Comments**

One major concern with the manuscript is the drawing of broad conclusions with limited samples. Specifically, connecting everything back to the biological content without knowledge of sample to sample variability during the different seasons. Given the samples collected this might not be possible, but some discussion of the degree of variability that would occur between two samples of the same season would help provide context as to how much these differences are due to real differences between seasons or differences that happen to occur between these two specific samples that could be due to other reasons.

The second concern is that, while the authors do go to great lengths to justify it, I remain not fully convinced that the organic spectra can be so definitively identified as alanine. Additionally, the confidence of assigning it to the Mg-hydrate salt seems not fully justified, given the challenge of distinguishing ions such as $Ca^{2+}$ from $Mg^{2+}$ in ambient vibrational spectra. I believe that alanine is likely one of a number of compounds contributing to the modes observed, along with the associated hydrate salts of different cations. The authors should soften that language a little to make clear that alanine is unlikely to be the sole contributor to those modes. The classification for fatty acids by contrast appears to be the appropriate level of specificity.

There is very little discussion of the nitrate observed in the samples, particularly in the sample S2. Given the distinctly different HYSPLITs between samples 1 and 2 the enrichment of $NO_3^-$ in S2 that passes over Chile seems to provide evidence of heterogeneous processing through the classic $NaCl + HNO_3 \rightarrow HCl + NaNO_3$ reaction. This is barely discussed, but seems like an important point that should be discussed in more detail. Aging of SSA and organics from SSA is well established with EDX and Raman [*Adachi and Buseck*, 2015; *Ault et al.*, 2013; *Ault et al.*, 2014; *Laskin et al.*, 2012; *Liu et al.*, 2011; *Trueblood et al.*, 2016], but primarily from CAICE-style waveflume experiments or in field studies the mid-latitudes. Discussion of heterogeneous aging in the less-studied Antarctic in the context of the impact observed and modeled [*Bauer et al.*, 2007] in the midlatitudes would strengthen the impact of the paper.

**Minor Comments**

This is a stylistic point, but I believe the "a" in "Chlorophyll-a" should be italicized.

The uncertainty of the chl-a measurements should be discussed further. Specifically, what is the LOD? The uncertainty should be included when referenced in the abstract and experimental sections in the form of +/- after the values. How certain are the authors that the difference is really 19 time for summer versus winter. This should be addressed further.

On lines 74-75 it could be interpreted that two samples were collected total or that 2 samples from summer and 2 from winter were collected. Consider revising to make this clear.

Lines 102-107 As noted above, some further comparison of the differences between the Hysplit trajectories would be useful for data interpretation.

Further discussion of the organic overlay from Raman and with elemental overlay from EDX would be useful. It seems that the organic portions from the Raman are the thickest portions. This is not necessarily intuitive, though the 2800-3000 window seems like a logical choice.

The silicon-containing particles are interesting. Why would the enhanced sea ice noted on lines 264-265 specifically hinder Si species emission? Is there something about the winter time emissions that is somehow chemically selective? This was unclear and should be explained in more detail.

For the 1052 cm$^{-1}$ peak noted on line 300, how are the authors able to distinguish $Mg(NO_3)_2$ the symmetric stretch of the aqueous nitrate ion observed around 1055 cm$^{-1}$? See [*Zangmeister and Pemberton*, 2001]and [*Ault et al.*, 2014] for reference. It seems unlikely that the two peaks could be distinguished cleanly and that even if one is present there may be contribution from the other mode.

The phrase "encountering frequency" is used frequently, but is an odd choice of wording. It is suggested that it be replaced with different terminology, as well as to clearly define the terminology used.

References:

Adachi, K., and P. R. Buseck (2015), Changes in shape and composition of sea-salt particles upon aging in an urban atmosphere, *Atmospheric Environment*, *100*, 1-9.
Ault, A. P., T. L. Guasco, O. S. Ryder, J. Baltrusaitis, L. A. Cuadra-Rodriguez, D. B. Collins, M. J. Ruppel, T. H. Bertram, K. A. Prather, and V. H. Grassian (2013), Inside versus outside: ion redistribution in nitric acid reacted sea spray aerosol particles as determined by single particle analysis, *Journal of the American Chemical Society*, *135*(39), 14528-14531.
Ault, A. P., T. L. Guasco, J. Baltrusaitis, O. S. Ryder, J. V. Trueblood, D. B. Collins, M. J. Ruppel, L. A. Cuadra-Rodriguez, K. A. Prather, and V. H. Grassian (2014), Heterogeneous reactivity of nitric acid with nascent sea spray aerosol: large differences observed between and within individual particles, *The Journal of Physical Chemistry Letters*, 2493-2500.
Bauer, S. E., D. Koch, N. Unger, S. M. Metzger, D. T. Shindell, and D. G. Streets (2007), Nitrate aerosols today and in 2030: a global simulation including aerosols and tropospheric ozone, *Atmospheric Chemistry And Physics*, *7*(19), 5043-5059.
Laskin, A., R. C. Moffet, M. K. Gilles, J. D. Fast, R. A. Zaveri, B. B. Wang, P. Nigge, and J. Shutthanandan (2012), Tropospheric chemistry of internally mixed sea salt and organic particles: Surprising reactivity of NaCl with weak organic acids, *Journal Of Geophysical Research-Atmospheres*, *117*, D017743.

Liu, Y., B. Minofar, Y. Desyaterik, E. Dames, Z. Zhu, J. P. Cain, R. J. Hopkins, M. K. Gilles, H. Wang, P. Jungwirth, and A. Laskin (2011), Internal structure, hygroscopic and reactive properties of mixed sodium methanesulfonate-sodium chloride particles, *Physical Chemistry Chemical Physics*, *13*(25), 11846-11857.

Trueblood, J. V., A. D. Estillore, C. Lee, J. A. Dowling, K. A. Prather, and V. H. Grassian (2016), Heterogeneous Chemistry of Lipopolysaccharides with Gas-Phase Nitric Acid: Reactive Sites and Reaction Pathways, *The Journal of Physical Chemistry A*, *120*(32), 6444-6450.

Zangmeister, C. D., and J. E. Pemberton (2001), Raman spectroscopy of the reaction of sodium chloride with nitric acid: Sodium nitrate growth and effect of water exposure, *Journal Of Physical Chemistry A*, *105*(15), 3788-3795.

---

## Author Comment (AC1) · 13 Oct 2016

**Comments (in *italics*) and responses**

**General Comments from Anonymous referee # 1**

*This manuscript is a very solid description of aerosols from a unique and understudied region (Antarctic). The study provides unique and valuable data regarding two samples (summer and winter) that have distinct differences in their composition. The combination of the three methods, particularly the Raman/ATR-FTIR comparison, is a powerful unified approach to explore the detailed physicochemical composition of these particles. The Ro group has really advanced the ATR-FTIR coupled with Raman approach that allows for insights regarding how sulfate is bound, for example. Overall, the data interpretation is thorough and solid. The authors do a nice job of placing their results in the context of other research in the field. A few small issues should be addressed in a revised manuscript and are indicated below. Overall, this manuscript has the potential to be a useful contribution to the field.*

**Response:** We thank the reviewer for his/her positive evaluation of our work and valuable comments.

**Major comments from Anonymous referee #1**

*One major concern with the manuscript is the drawing of broad conclusions with limited samples. Specifically, connecting everything back to the biological content without knowledge of sample to sample variability during the different seasons. Given the samples collected this might not be possible, but some discussion of the degree of variability that would occur between two samples of the same season would help provide context as to how much these differences are due to real differences between seasons or differences that happen to occur between these two specific samples that could be due to other reasons.*

**Response:** The reviewer is right in terms that it is unclear whether the different compositional features between two samples are due to their different sampling seasons or biological activities. To answer this question, a study for overall 29 Antarctic aerosol samples is underway. We hope that this further study can provide us better understanding about the Antarctic SSAs. In the

revised manuscript, we mentioned this point by modifying a paragraph ("First, although just two SSA samples, having a high chlorophyll-*a* level contrast, collected in the summer and winter were investigated, their chemical compositional features were clearly different in terms of their chemical species and/or levels of inorganic and organic moieties. However, it is not clear whether the different compositional features are due to their different sampling seasons or biological activities. To answer this question, a study for overall 29 Antarctic aerosol samples collected during Dec. 2011 and Sep. 2012 when the oceanic chlorophyll-*a* levels were in the range of 0.07-13.38 μg/L is underway." – p. 18, lines 528-535 in the revised manuscript).

*The second concern is that, while the authors do go to great lengths to justify it, I remain not fully convinced that the organic spectra can be so definitively identified as alanine. Additionally, the confidence of assigning it to the Mg-hydrate salt seems not fully justified, given the challenge of distinguishing ions such as $Ca^{2+}$ from $Mg^{2+}$ in ambient vibrational spectra. I believe that alanine is likely one of a number of compounds contributing to the modes observed, along with the associated hydrate salts of different cations. The authors should soften that language a little to make clear that alanine is unlikely to be the sole contributor to those modes. The classification for fatty acids by contrast appears to be the appropriate level of specificity.*

**Response:** We admit that Raman and ATR-FTIR spectra of MgAla-containing Antarctic SSAs do not perfectly match with those of aerosols nebulized from standard alanine and MgCl2 mixture solutions. Therefore, MgAla-containing SSAs should be mainly composed of MgAla with the very small amounts of other unknown components. It is also true that sometimes Ca salts cannot be easily distinguished from Mg salts, and yet most Ca ions are combined with sulfate according to X-ray and Raman mapping results, excluding the consideration of Ca salts in the assignment. We carefully rewrote the revised manuscript not to convey the impression that MgAla-containing SSAs are solely composed of MgAla. Regarding this comment, some modification is given in the revised manuscript ("the frequently observed organic species are most probably ones containing Mg hydrate salts of alanine (MgAla) and Mg salts of fatty acids (MgFAs)." – p. 11, lines 304-305; "Divalent Ca ions are also present in sea-water. However, based on X-ray and Raman mapping results, Ca ions are mostly combined with inorganic $SO_4^{2-}$ and slightly present in regions where organic moieties are. Based on a comparison of the Raman and ATR-FTIR spectra obtained for the summertime SSAs and

aerosols generated from the mixture solution of standard alanine and $MgCl_2$, the organic species are ones containing mainly the Mg hydrate salts of alanine (MgAla), even though their precise molecular formula and the other possible minor components could not be confirmed." – p.12, lines 353-360; "it is clear that MgAla-containing SSAs are composed of mainly alanine with negligible glycine and other target chemicals." – p.13, line 383-384) together with tone-downed wording.

*There is very little discussion of the nitrate observed in the samples, particularly in the sample S2. Given the distinctly different HYSPLITs between samples 1 and 2 the enrichment of $NO_3^-$ in S2 that passes over Chile seems to provide evidence of heterogeneous processing through the classic $NaCl + HNO_3 \rightarrow HCl + NaNO_3$ reaction. This is barely discussed, but seems like an important point that should be discussed in more detail. Aging of SSA and organics from SSA is well established with EDX and Raman [Adachi and Buseck, 2015; Ault et al., 2013; Ault et al., 2014; Laskin et al., 2012; Liu et al., 2011; Trueblood et al.,2016], but primarily from CAICE-style waveflume experiments or in field studies the mid-latitudes. Discussion of heterogeneous aging in the less-studied Antarctic in the context of the impact observed and modeled [Bauer et al., 2007] in the midlatitudes would strengthen the impact of the paper.*

**Response:** NO3- is more frequently observed in sample S1 than in sample S2 (see Table 2), suggesting nitrates of a marine origin rather than of an anthropogenic origin from Chile. To discuss the observation of nitrates, we insert a paragraph ("Although N X-ray signal was not detected probably due to the small amount of $NO_3^-$ present in the Antarctic SSAs, $Mg(NO_3)_2$ and $NaNO_3$ were frequently observed in samples S1 and S2 using Raman and ATR-FTIR techniques. The nitrate in sea-water can be generated by the photoammonification process, which transforms dissolved organic nitrogen (DON) to labile inorganic nitrogen, mainly ammonium ($NH_4^+$) (Kitidis et al., 2006; Aarnos et al., 2012; Xie et al., 2012; Rain-Franco et al., 2014; Paulot et al., 2015), followed by the microbial oxidation of ammonium into nitrate ($NO_3^-$) by nitrifying bacteria (Carlucci et al., 1970; Hovanec and Delong, 1996; Smith et al., 2014; Tolar et al., 2016). As the photoammonification depends on solar radiations, the ammonium and nitrate production would be enhanced in the summer with higher solar radiation level. Indeed, as shown in Table 2, nitrates are more frequently observed in summertime sample S1 than wintertime sample S2." – p. 15, lines 439-449).

**Minor comments from Anonymous referee #1**

*This is a stylistic point, but I believe the "a" in "Chlorophyll-a" should be italicized.*

**Response:** Corrected in revised text. We thank the reviewer for pointing out the error.

*The uncertainty of the chl-a measurements should be discussed further. Specifically, what is the LOD? The uncertainty should be included when referenced in the abstract and experimental sections in the form of +/- after the values. How certain are the authors that the difference is really 19 time for summer versus winter. This should be addressed further.*

**Response:** We did not clearly state that the *chl-a* levels are for the nearby oceanic water collected on the sampling days of the samples, which are not mean *chl-a* levels for the summer and winter. In the revised manuscript, we made this point clear and provided the description of *chl-a* determination and references ("This study examined two aerosol samples S1 and S2 collected on Dec. 9, 2011 in the austral summer and July 23, 2012 in the austral winter, respectively, when the oceanic chlorophyll-*a* levels on the collection days of the samples were quite different, by ~19 times (2.46 vs. 0.13 μg/L, respectively). The oceanic chlorophyll-*a* levels for water samples collected daily from nearby oceanic water were determined using a fluorometer (TD 700, Turner Design, USA)). The detailed description for chlorophyll-*a* determination is given elsewhere (Schloss et al., 2014; Lee et al., 2015)." – p. 4, lines 93-99).

*On lines 74-75 it could be interpreted that two samples were collected total or that 2 samples from summer and 2 from winter were collected. Consider revising to make this clear.*

**Response:** Corrected ("two Antarctic aerosol samples collected on Dec. 9, 2011 in the austral summer (sample S1) and July 23, 2012 in the austral winter (sample S2)" p. 3, lines 75-77).

*Lines 102-107 As noted above, some further comparison of the differences between the Hysplit trajectories would be useful for data interpretation.*

**Response:** Please refer to our response above.

*Further discussion of the organic overlay from Raman and with elemental overlay from EDX would be useful. It seems that the organic portions from the Raman are the thickest portions. This is not necessarily intuitive, though the 2800-3000 window seems like a logical choice.*

**Response:** Further discussion is given in the revised manuscript **(**"Molecular Raman images look broader than elemental X-ray images as the spatial resolution of Raman mapping (~1 μm) is larger than that of X-ray mapping (~0.1 μm). Especially, Raman images for organic species look more spread than C X-ray map images as the low energy C X-rays generated from underneath are not often detected due to the strong absorption by solid particles sitting above." – p. 7, lines 198-202).

*The silicon-containing particles are interesting. Why would the enhanced sea ice noted on lines 264-265 specifically hinder Si species emission? Is there something about the winter time emissions that is somehow chemically selective? This was unclear and should be explained in more detail.*

**Response:** Production of silicone component can be related to diatom activities, which would be higher in the summer. As the effect of sea ice cap is not clear, the ice cap thing is removed in the revised manuscript ("In the winter, the reduced diatom activities would decrease the emission of Si species into the atmosphere, resulting in the scarce observation of Si in the S2 sample." – p. 8, lines 274-275).

*For the 1052 $cm^{-1}$ peak noted on line 300, how are the authors able to distinguish $Mg(NO_3)_2$ the symmetric stretch of the aqueous nitrate ion observed around 1055 $cm^{-1}$? See [Zangmeister and Pemberton, 2001]and [Ault et al., 2014] for reference. It seems unlikely that the two peaks could be distinguished cleanly and that even if one is present there may be contribution from the other mode.*

**Response:** As Raman spectra show no free water in the SSAs and SEIs indicate solid SSAs, we think that the presence of aqueous nitrate is ignorable.

*The phrase "encountering frequency" is used frequently, but is an odd choice of wording. It is suggested that it be replaced with different terminology, as well as to clearly define the terminology used.*

**Response:** We agree with the reviewer's comment. In the revised manuscript, "encountered" is mostly replaced by "observed". The definition of "encountering frequency" is given in the revised manuscript ("where the relative encountering frequency (in %) for a certain element is defined as the number of particles containing the element divided by the total number of particles analyzed for a sample" – p. 6, lines 176-178).

---

## Author Comment (AC2) · 13 Oct 2016

**Comments (in *italics*) and responses**

**General Comments from Anonymous referee # 2**

*This is an interesting paper that focuses on the integration of several single particle techniques to investigate sea spray aerosol samples collected at King Sejong Korean scientific research station in the austral summer and in the austral winter. Through use of these single particle techniques several conclusions are drawn related to the identification of organic compounds and inorganic salts. Some of the observations reported in the paper have been seen before and confirm earlier studies (e.g. chloride depletion in particles).*

**Response:** We thank the reviewer for the positive evaluation of our work.

**Specific comments from Anonymous referee # 2**

*In many particles, there is nitrate observed in the particles - where does the nitrate come from in these particles?*

**Response:** We provided discussion in the revised manuscript as follows; "Although N X-ray signal was not detected probably due to the small amount of $NO_3^-$ present in the Antarctic SSAs, $Mg(NO_3)_2$ and $NaNO_3$ were frequently observed in samples S1 and S2 using Raman and ATR-FTIR techniques. The nitrate in sea-water can be generated by the photoammonification process, which transforms dissolved organic nitrogen (DON) to labile inorganic nitrogen, mainly ammonium ($NH_4^+$) (Kitidis et al., 2006; Aarnos et al., 2012; Xie et al., 2012; Rain-Franco et al., 2014; Paulot et al., 2015), followed by the microbial oxidation of ammonium into nitrate ($NO_3^-$) by nitrifying bacteria (Carlucci et al., 1970; Hovanec and Delong, 1996; Smith et al., 2014; Tolar et al., 2016). As the photoammonification depends on solar radiations, the ammonium and nitrate production would be enhanced in the summer with higher solar radiation level. Indeed, as shown in Table 2, nitrates are more frequently observed in summertime sample S1 than wintertime sample S2." – p. 15, lines 439-449.

*Why is alanine such a dominant factor in the SSA? Are there other compounds that can have similar spectral features? It seems too simple to have one compound and one complex Mg-alanine in sea spray particles. The case for fatty acids seems more convincing given several studies that have identified palmitate and stearate in sea spray aerosol.*

**Response:** The reviewer #1 also made the same comment. Please refer to our response to the comment by the reviewer #1.

*Some of the figure captions in the supplemental do not match or explain well the figure making it difficult to understand what is being shown. (See for example Figure S3 – what are the three sets of spectra shown?)*

**Response:** Figure captions in the supplement are modified as below.

Figure S3. Raman and ATR-FTIR spectra of the aerosols generated by the nebulization of a mixture solution of 0.2 M alanine and 0.1 M $MgCl_2$ standard chemicals. The first pair of Raman and ATR-FTIR spectra for the aerosols was obtained just after the nebulization and the second and third pairs of Raman and ATR-FTIR spectra were obtained ~1 year later after the storage in a desiccator. The first and third pairs of Raman and ATR-FTIR spectra for organic moiety look similar to those in Figures 4(b) and 4(a), respectively

Figure S4. Raman and ATR-FTIR spectra of some target chemicals for organics in Antarctic SSAs, which do not resemble with those for MgAla-containing SSAs.

Figure S5. Raman and ATR-FTIR spectra of powdery standard Mg palmitate, palmitic acid, Mg stearate, and stearic acid, which are sufficiently different to distinguish the four compounds

Figure S6. ATR-FTIR spectra of Mg palmitate, Mg stearate, a mixture of Mg palmitate and stearate (by 3:1), and MgFAs-containing SSA, showing that MgFAs-containing SSAs are the mixture of mainly Mg palmitate and stearate.

Figure S7. Raman and ATR-FTIR spectra of standard inorganic chemicals, which are observed in Antarctic SSAs.